# Use of Cabbage Leaf Inverted Flap Technique in the Management of a Stage IV Full-Thickness Macular Hole

**DOI:** 10.3390/jcm13237120

**Published:** 2024-11-25

**Authors:** Kristina J. Hartung, Fran Drnovšek, Xhevat Lumi

**Affiliations:** 1Department of Ophthalmology, University Medical Centre Ljubljana, 1000 Ljubljana, Slovenia; kristina.j.hartung@kclj.si (K.J.H.); fran.drnovsek@gmail.com (F.D.); 2Faculty of Medicine, University of Ljubljana, 1000 Ljubljana, Slovenia

**Keywords:** full-thickness macular hole, pars plana vitrectomy, cabbage leaf technique, hole closure, visual acuity

## Abstract

**Background:** The purpose of this study was to evaluate the anatomical and visual outcomes of patients with stage IV full-thickness macular holes (FTMHs) treated by the cabbage leaf inverted internal limiting membrane (ILM) flap technique. **Methods:** We conducted a retrospective study, enrolling patients with stage IV FTMH operated by a single surgeon. Six patients with FTMH and no other known ocular comorbidities were included in the study. **Results:** Four patients (66.6%) were female, two were male. The median age was 71. The average duration of symptoms before surgery was 10.5 months (6–24 months). The mean preoperative minimum hole diameter was 480 μm (337–602), and the mean basal hole size was 1208 μm (703–1748). The mean preoperative BCVA was 0.63 LogMAR. Postoperatively, the BCVA improved in 5 (83.3%) patients and remained the same in 1 (16.7%). The mean postoperative BCVA was 0.42 LogMAR (0.0–0.70). The FTMH has closed in all cases (100%). At the follow-up examination three months after the surgery, we observed complete closure of the FTMH with the restoration of retinal layers. **Conclusions:** This approach resulted in a complete closure of FTMH with significant visual acuity improvement. The technique could represent the surgical procedure of choice in the management of stage IV FTMH.

## 1. Introduction

Full-thickness macular hole (FTMH) is a foveal lesion, defined as a full-thickness neurosensory retinal defect in the central fovea, which is an important cause of central visual loss [1,2]. Primary macular hole results from vitreomacular traction, while secondary macular hole occurs due to other pathologic conditions, such as trauma or ruptured foveal cystoid spaces in chronic macular edema, laser treatment, diabetic retinopathy, ruptured retinal arterial macroaneurysm, and vitelliform dystrophy, and does not have a component of vitreomacular traction [2]. The first macular hole (MH) classification was proposed by Gass, and MH was subclassified into four groups based on their clinical appearance [1]. Stage 1 MH, also known as impending macular hole, involves the loss of the foveal depression, with two subtypes: Stage 1A, which shows foveolar detachment with a loss of foveal contour and a lipofuscin-colored spot, and Stage 1B, where the detachment is marked by a lipofuscin-colored ring. Stage 2 MH is characterized by a full-thickness retinal break less than 400 µm in size, often with a preserved inner layer “roof.” This stage typically occurs within weeks to months after Stage 1 and is associated with a decline in visual acuity. Stage 3 MH represents further progression, with the hole expanding to a size greater than 400 µm. A grayish rim and subretinal fluid often appear, and the posterior hyaloid detachment becomes evident, sometimes with an operculum (a flap of retinal tissue). Stage 4 MH occurs when Stage 3 is accompanied by a complete posterior vitreous detachment, marked by a Weiss ring. As the hole advances, visual acuity typically deteriorates, with a significant decline in central vision due to retinal tissue disruption [1]. Nevertheless, the Gass classification has been gradually replaced by a grading system based on the FTMH size as measured on the optical coherence tomography (OCT) proposed by The International Vitreomacular Traction Study Group [1]. They have anatomically divided macular holes based on the aperture size into small (diameter < 250 μm), medium (diameter from 250 to 400 μm), and large (diameter > 400 μm) [1]. Furthermore, the presence of vitreous detachment is important in the subcategorization of macular holes, and large macular holes with posterior vitreous detachment have been defined as stage IV [1]. This grading system allows for a more standardized approach in the evaluation of macular holes, which is essential for guiding treatment decisions and predicting outcomes. A classification based on the duration of the macular hole as well as the symptoms has also been proposed. Bamberger et al. proposed classifying FTMH as chronic if the patients reported a history of symptoms for over 2 years or a 1-year duration of FTMH after diagnosis [2]. Several surgical techniques have been used in the management of FTMH. Pars plana vitrectomy (PPV) with peeling of the internal limiting membrane (ILM) is the most often used technique in the management of FTMH [3]. However, stage IV FTMHs are known to have lower rates of anatomical closure with ILM peel and poor functional recovery even when hole closure is achieved [3]. Therefore, the inverted ILM flap technique has become a mainstay in the treatment of stage IV FTMH [4,5]. It has been proposed that the ILM flap creates a favorable milieu for the migration of neurosensory cells and glial proliferation. It also serves as a scaffold for migrating cells, resulting in FTMH closure and visual acuity improvement [4,5,6]. Even though higher anatomical success rates were reported with the inverted flap technique, in a substantial number of cases, it still fails to deliver anatomical and functional improvement [4,6,7]. Therefore, several modifications of the technique have been suggested. Among those, Aurora et al. described a technique of multiple ILM flaps [8]. Rather than a circular ILM peel, several separate flaps are formed, with their base attached at the edge of FTMH. ILM flaps are inverted over each other and over the hole in a manner resembling cabbage leaves. The interlocking of the flaps prevents their displacement during the fluid–air exchange and establishes reliable coverage of the hole. The efficacy of this technique was also demonstrated in the treatment of FTMH coexisting with rhegmatogenous retinal detachment in high myopic eyes [9]. We present a case series of stage IV FTMH managed by the cabbage leaf inverted ILM flap technique.

## 2. Materials and Methods

We conducted a retrospective study, enrolling patients with stage IV FTMH, operated by a single senior consultant surgeon at the Department of Ophthalmology University Medical Centre Ljubljana, Ljubljana, Slovenia. This study was conducted according to the guidelines of the Declaration of Helsinki and approved by the National Medical Ethics Committee of the Republic of Slovenia (Approval No. 91/05/11). Patients were enrolled from April 2021 to May 2023. Inclusion criteria were as follows: Patients with stage IV macular holes and pseudophakia who underwent surgery with the cabbage leaf technique. Exclusion criteria were preexisting ocular pathology (e.g., age-related macular degeneration, glaucoma, high myopia (>−6), uveitis) and concomitant systemic comorbidities (e.g., diabetes). The patient’s medical records were reviewed, and the following data were retrieved: age, gender, symptom duration, preoperative best-corrected visual acuity (BCVA), and postoperative BCVA at 3 months post-surgery. Optical coherence tomography images were obtained using the swept-source OCT (SS-OCT), Topcon DRI OCT Triton (Topcon Corp., Tokyo, Japan) before surgery and at a follow-up at 3 months post-surgery. PPV using a 25-gauge system (Constellation, Alcon, Fort Worth, TX, USA) was performed. At least two separate ILM flaps with their base connected to the macular hole margin were made during the ILM peel. Flaps were trimmed with a vitreous cutter and gently inverted over the hole and each other in a cabbage leaf-like manner (Figure 1, Appendix A). As a rule, one flap was made temporally and the other nasally from the FTMH; we occasionally also made a third flap above or below the FTMH. After the fluid–air exchange, a gas tamponade using 10% of C3F8 was instilled. The patients were advised to maintain a head-down position for at least 3 days following the surgery. All patients were followed up at 3 months after the surgery.

## 3. Results

Six patients with FTMH and no other known ocular comorbidities were included in the study. Demographic and clinical characteristics are presented in Table 1. Four (66.6%) were female, and two were males (33.4%). Median age was 71 years (56–84). The mean duration of symptoms, including diminished visual acuity and metamorphopsia before surgery, was 10.5 months (3–24 months). The mean preoperative minimum hole diameter was 480 μm (337–602), and the mean basal hole size was 1208 μm (703–1748). The preoperative BCVA was 0.7 logMAR in five and 0.3 logMAR in one patient. Postoperatively, the BCVA improved in 5 (83.3%) patients and remained the same in 1 (16.7%). The average postoperative BCVA was 0.42 LogMAR, ranging from 0.60 to 0.0. The patient with the best preoperative BCVA (0.3 LogMAR) exhibited the best postoperative BCVA (0.0 LogMAR). The FTMH has closed in all 6 cases (100%) (Figure 2). At the follow-up examination three months after the surgery, we observed complete closure of the FMMH, with the restoration of retinal layers that remained stable during the 1-year follow-up period (Figure 3). We did not find any intra- or postoperative complications in any of these cases.

## 4. Discussion

Surgical management of a stage IV FTMH is often challenging. In our experience, the cabbage leaf inverted flap technique resulted in a complete closure of a stage IV FMTH with significant visual function improvement. There are several advantages to the proposed technique. Firstly, two to three separate flaps form a complete and stable cover of the FTMH. The retinal surface of the first flap adheres to the vitreal surface of an overlaying flap, locking them together and preventing their displacement. Moreover, the flaps are only laid over the FTMH, minimizing the risk of touching and potentially damaging the RPE in the foveal region. It is crucial to maintain the flaps on top of the retina, covering the hole, as that provides a scaffold for the migration of the glial cells [5,10]. A closed space between the ILM flaps, retinal pigment epithelium (RPE), and neurosensory retina is formed, which is considered to promote the centripetal migration of the neurosensory retinal tissue, resulting in FTMH closure. Even though the exact pathophysiological mechanisms behind the closure of idiopathic FTMHs are yet unknown, a study using an experimental model on monkeys demonstrated that proteins integral to the ILM could promote the proliferation and migration of the Müller cells [5,10]. It also showed that migrating Müller cells express more neurotrophic factors (NF)—such as brain-derived NF, glial cell-derived NF, and ciliary NF—as well as basic fibroblast growth factor (bFGF) than stable, non-migrating Müller cells [5,10]. These factors are presumably involved in the FTMH closure and recovery of photoreceptors [5,10]. Furthermore, covering the hole with the ILM flap results in glial activation and Müller cell gliosis. Even though Müller cell gliosis seems to be essential in the healing of macular holes, severe gliosis has been associated with worse visual outcomes as it prevents the normal external limiting membrane (ELM) and photoreceptor alignment restoration [11]. Particularly, larger macular holes were associated with higher rates of severe glial proliferation, possibly preventing the restoration of photoreceptor alignment and resulting in worse visual outcomes. Glial proliferation at the photoreceptor layer is less profound with the inverted ILM flap technique as compared to the insertion technique, resulting in better restoration of the outer retinal layer [12]. A significant correlation was shown between the retinal outer layer restoration and the improvement of visual acuity [12]. This could explain the favorable visual outcome of the proposed technique in our case series. Even though several modifications of the original ILM inverted flap technique have been previously proposed, they come with certain limitations. The temporal inverted ILM flap aims to obtain FTMH closure with minimal iatrogenic trauma; however, the unfolding of the flap during fluid–air exchange can cause surgical failure, resulting in a poor anatomical and functional outcome, requiring revision surgery [6,7]. To address this problem, a large semicircular inverted ILM method was proposed [13], but the formation of a large ILM flap is challenging and, in some cases, unattainable. The C-shaped temporal inverted ILM flap technique proposed a smaller flap, 1.5-disc diameter in size, cut 270 degrees around the FTMH. However, the potential unfolding of the flap could again pose a problem [14]. In the proposed cabbage leaf technique, separate flaps form a complete and stable cover of the FTMH. Stability is achieved by overlapping the flaps. The retinal surface of the first flap adheres to the inner surface of an overlaying flap, locking them together and preventing their displacement. Smaller flaps can be created in a more controlled manner, decreasing the risk of iatrogenic damage. Moreover, the technique is also possible in the eyes, where the formation of a large ILM flap is not attainable, as it was in our case with the accompanying ERM within temporal vascular arcades. In addition, the flaps are only laid over the FTMH, minimizing the risk of touching and potentially damaging the retinal pigment epithelium (RPE) in the foveal region. This could help to explain the favorable anatomical and functional outcomes that we observed in our patients who underwent vitrectomy with the cabbage leaf technique.

It is essential to recognize the limitations of this study. Firstly, a rather small sample size resulted from the narrow inclusion criteria, focusing solely on patients with a stage IV macular hole who are also pseudophakic. In addition, this study does not compare the cabbage leaf technique with alternative surgical approaches, such as the internal limiting membrane (ILM) peel or the inverted flap technique, which is commonly employed in retinal surgeries for macular hole repair. Each of these methods has its own advantages and potential limitations, which may impact surgical outcomes such as closure rates, recovery time, and postoperative visual acuity. Comparative studies between the cabbage leaf technique and these alternatives would provide valuable insights into the relative effectiveness, safety profiles, and long-term outcomes of each approach.

## 5. Conclusions

The cabbage leaf technique resulted in a complete closure of a stage IV FTMH in all cases, with a significant improvement of visual function in most of the cases. The technique could represent the surgical procedure of choice in the management of stage IV macular holes. Further studies in a larger number of cases are needed to assess the method thoroughly.

## Figures and Tables

**Figure 1 jcm-13-07120-f001:**
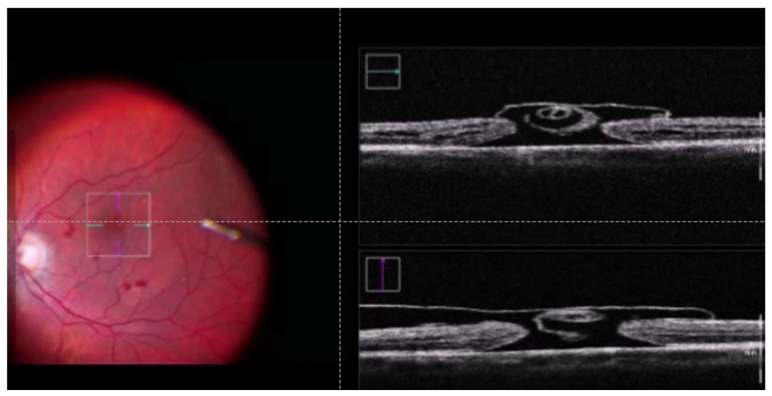
Intraoperative OCT exhibits the formation of ILM flaps resembling a cabbage leaf overlying the full-thickness macular hole.

**Figure 2 jcm-13-07120-f002:**
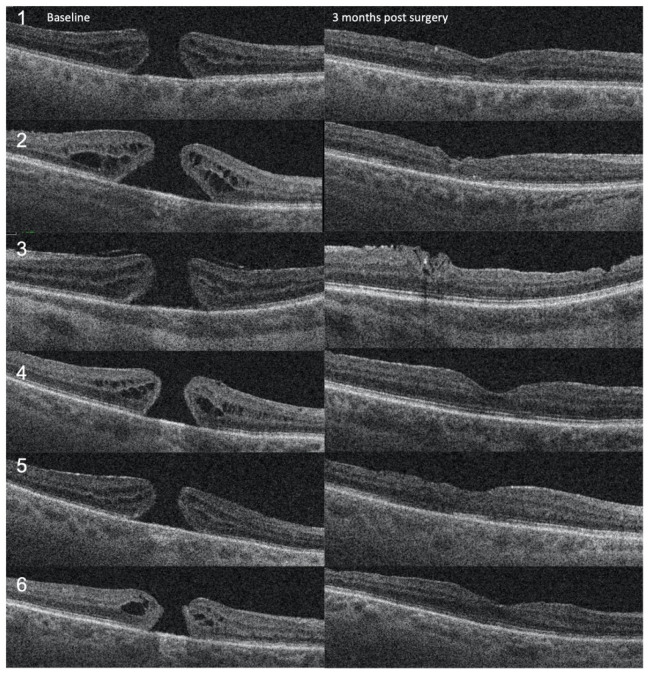
OCT follow-up of the 6 patients from baseline to 3 months post-surgery.

**Figure 3 jcm-13-07120-f003:**
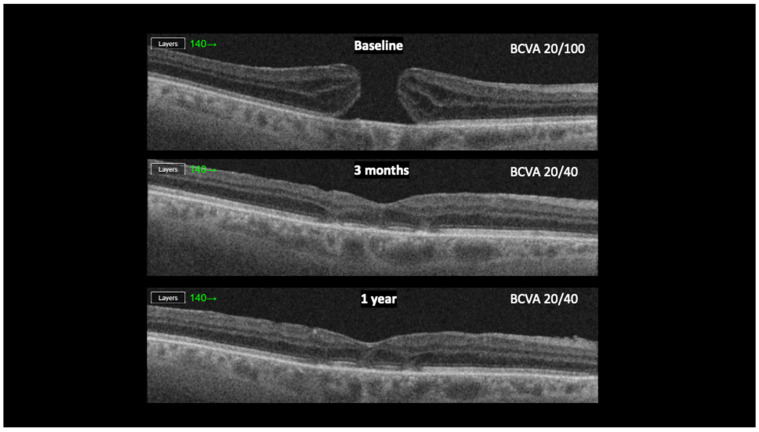
OCT follow-up of the patient from baseline to 1 year post-surgery.

**Table 1 jcm-13-07120-t001:** Demographic and clinical characteristics.

Pt.	Age	Gender	Duration of Symptoms	Minimum Hole Size (μm)	Basal Hole Size (μm)	Preoperative BCVA (LogMAR)	Postoperative BCVA (LogMAR)
1	71	M	18 months	549	1224	0.7	0.3
2	84	F	24 months	526	1748	0.7	0.5
3	56	F	6 months	602	1343	0.7	0.7
4	71	F	3 months	443	1063	0.3	0.0
5	73	M	6 months	424	1169	0.7	0.5
6	72	F	6 months	337	703	0.7	0.4

## Data Availability

Dataset available on request from the authors.

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
