# Peer review of "Use of Cabbage Leaf Inverted Flap Technique in the Management of a Stage IV Full-Thickness Macular Hole"

_jcm, 2024, doi:10.3390/jcm13237120_

Round 1

Reviewer 1 Report

Comments and Suggestions for Authors

Dear authors,

After reading your manuscript, i have the following suggestions to improve the impact that this paper may have:

Intro section

A short mention of the diagnostic criteria for FTMH is in order, mention of OCT and use this opportunity to introduce your abbreviations for OCT and SD OCT.

M&M section

The design is mentioned as being prospective, but your group have a 2022 case report of a 71 year old male that looks surprisingly similar to Case 1 in this case series. It's better to adequately report the study as being retrospective, if it indeed is so. You then go on to say " The patient’s medical records were reviewed"; this implies a retrospective design.

Also, you should mention the dates when recruiting began and when it ended.

There is inadequate mention of the exact timing of the post-op visit where BCVA was determined.

There is no mention of the reproducibility of the cabbage flap technique. Is this something that can be reliably applied to each case? If so, maybe some technical tips in the M&M section could be added. If not, then specify when and where.

Results section

The first sentence in the results section should read "Six patients ....", not "6 patients..."

Row 71, ", in all cases a large, FTMH was present", no need for a comma here

Row 72,  Optical coherence tomography (SS-72 OCT, Topcon DRI OCT Triton; Topcon Corp., Tokyo, Japan). This belongs in the M&M section and it already is there so no need for the parenthesis.

Also, there is no need to say that the FTMH was present pre-op because this is your inclusion criteria, of course it is present.

Row 80, At the follow-up examination three months after the surgery. You should mention in the M&M section that the follow-up was performed at 3 months for all patients. And if this is the case for patient 1 as well, check that 2022 abstract from EuRetina where the post-op BCVA was 0.5 decimal (0.3 logMAR). Also, correct the BCVA in the table from 0.20 to 0.2

Conclusions are adequate

Author Response

Dear authors,

After reading your manuscript, i have the following suggestions to improve the impact that this paper may have:

Thank you so much for taking the time to review our paper and your thorough and generally favorable review.

Intro section

A short mention of the diagnostic criteria for FTMH is in order, mention of OCT and use this opportunity to introduce your abbreviations for OCT and SD OCT.

Response: Thank you for your suggestion. We have updated the section as follows:

Full-thickness macular hole (FTMH) is defined as full-thickness neurosensory retinal defect in the central fovea which is an important cause of central visual loss [1, 2]. Primary macular hole results from vitreomacular traction, while secondary macular hole occurs due to other pathologic conditions, such as trauma or foveal cystoid spaces in chronic macular odema, laser treatment, diabetic retinopathy, ruptured retinal arterial macronaeuysm, viteliform dystrophy and does not have a component of vitreomacular traction [2]. The International Vitreomacular Traction Study Group has proposed a grading system based on its size on optical coherence tomography (OCT) [2]. According to that system, macular holes are classified based on the aperture size into small (diameter < 250 μm), medium (diameter from 250 to 400  μm), and large (diameter > 400  μm) [2]. Furthermore, the presence of vitreous detachment was recognised as important in the subcategorization of macular holes, and large macular holes with posterior vitreous detachment have been defined as stage IV [2]. A classification based on the duration of the macular hole as well as the symptoms has also been proposed. Bamberger et al. proposed classifying FTMH as chronic if the patients reported a history of symptoms for over 2 years or a 1-year duration of FTMH after diagnosis [3].

M&M section

The design is mentioned as being prospective, but your group have a 2022 case report of a 71 year old male that looks surprisingly similar to Case 1 in this case series. It's better to adequately report the study as being retrospective, if it indeed is so. You then go on to say " The patient’s medical records were reviewed"; this implies a retrospective design. Also, you should mention the dates when recruiting began and when it ended.

Response: Thank you for pointing this out; the study was indeed retrospective. We have corrected the manuscript accordingly. And have entered the dates of recruiting as follows:

»We conducted a retrospective study, enrolling patients with chronic stage IV FTMH, operated by a single senior consultant surgeon at the Department of Ophthalmology University Medical Centre Ljubljana, Ljubljana, Slovenia. Patients were enrolled from April 2021 to May 2023.«

There is inadequate mention of the exact timing of the post-op visit where BCVA was determined.

Response: We have updated the manuscript accordingly:

The patient’s medical records were reviewed, and the following data were retrieved: age, gender, symptom duration, preoperative best-corrected visual acuity (BCVA), and postoperative BCVA at 3 months post-surgery. 

There is no mention of the reproducibility of the cabbage flap technique. Is this something that can be reliably applied to each case? If so, maybe some technical tips in the M&M section could be added. If not, then specify when and where.

Response: We have updated the manuscript and added a video clip showing the technique.

»At least two separate flaps with their base connected to the macular hole margin were made during the ILM peel. Flaps were trimmed with a vitreous cutter and gently inverted over the hole and each other in a cabbage leaf-like manner (Figure 1, Video 1).«

Results section

The first sentence in the results section should read "Six patients ....", not "6 patients..."

Response: Thank you for pointing this out, we have corrected it: »Six patients with FTMH and no other known ocular comorbidities were included in the study.«

Row 71, ", in all cases a large, FTMH was present", no need for a comma here. Row 72,  Optical coherence tomography (SS-72 OCT, Topcon DRI OCT Triton; Topcon Corp., Tokyo, Japan). This belongs in the M&M section and it already is there so no need for the parenthesis. Also, there is no need to say that the FTMH was present pre-op because this is your inclusion criteria, of course it is present.

Response: Thank you for pointing this out, we have removed the pararaph.

Row 80, At the follow-up examination three months after the surgery. You should mention in the M&M section that the follow-up was performed at 3 months for all patients. And if this is the case for patient 1 as well, check that 2022 abstract from EuRetina where the post-op BCVA was 0.5 decimal (0.3 logMAR). Also, correct the BCVA in the table from 0.20 to 0.2

We have updated the MM section as follows:

»The patient’s medical records were reviewed, and the following data were retrieved: age, gender, symptom duration, preoperative best-corrected visual acuity (BCVA), and postoperative BCVA at 3 months post-surgery.#

That is very good observation; this is indeed the same patient. We have corrected the table accordingly.

Conclusions are adequate

Reviewer 2 Report

Comments and Suggestions for Authors

Although this paper aims to report on the effectiveness of the cabbage leaf inverted ILM flap technique for chronic macular holes, it includes cases that are not chronic (i.e. patient 4 with a symptom duration of 3 months). A recent paper has defined chronic macular holes as those with > 1-year duration after diagnosis or those with symptoms for > 2 years' duration.1 The definition of chronic macular holes should be clearly stated in the paper based on previous reports. Personally, I think the study might be better if it only included Patients 1 and 2.

1.         Bamberger MD, Felfeli T, Politis M, Mandelcorn ED, Galic IJ, Chen JC. Human Amniotic Membrane Plug for Chronic or Persistent Macular Holes. Ophthalmol Retina. 2022 May;6(5):431-433. doi: 10.1016/j.oret.2022.01.006. Epub 2022 Jan 17. PMID: 35051667.

Was this study approved by the ethics committee? If this is a prospective study, was the definition of chronic macular holes established at that time? If not, it would be better to restructure this as a retrospective study targeting chronic macular holes.

Please consider including the inclusion and exclusion criteria in the paper.

Please provide more detailed information about the surgical procedure, such as: Was the ILM flap used to cover the macular hole or was it inserted into the hole? Were patients instructed to maintain prone positioning postoperatively, and if so, for how many days?

The limitations of the study should be discussed, such as this being a case series and the lack of comparison with other surgical techniques.

Author Response

Although this paper aims to report on the effectiveness of the cabbage leaf inverted ILM flap technique for chronic macular holes, it includes cases that are not chronic (i.e. patient 4 with a symptom duration of 3 months). A recent paper has defined chronic macular holes as those with > 1-year duration after diagnosis or those with symptoms for > 2 years' duration.1 The definition of chronic macular holes should be clearly stated in the paper based on previous reports. Personally, I think the study might be better if it only included Patients 1 and 2.

  1. Bamberger MD, Felfeli T, Politis M, Mandelcorn ED, Galic IJ, Chen JC. Human Amniotic Membrane Plug for Chronic or Persistent Macular Holes. Ophthalmol Retina. 2022 May;6(5):431-433. doi: 10.1016/j.oret.2022.01.006. Epub 2022 Jan 17. PMID: 35051667.

Thank you for pointing this out. We have updated the introduction accordingly and changed the title from chronic to stage IV.

»Full-thickness macular hole (FTMH) is a foveal lesion, defined as the absence of all retinal layers from the internal limiting membrane to the retinal pigment epithelium, which is an important cause of central visual loss [1, 2]. Primary macular hole results from vitreomacular traction while secondary macular hole occurs due to other mechanical causes, such as trauma or tractional foveal cystoid spaces in chronic macular odema, and does not have a compontent of vitreomacular traction [2]. The International Vitreomacular Traction Study Group has proposed a grading system based on its size on optical coherence tomography (OCT) [2]. They have anatomically divided macular holes based on the aperture size into small (diameter < 250 μm), medium(diameter from 250 to 400  μm), and large (diameter > 400  μm) [2]. Furthermore, the presence of vitreous detachment is important in the subcategorization of macular holes,  and large macular holes with posterior vitreous detachment have been defined as stage IV [2].  A classification based on the duration of the macular hole as well as the symptoms has also been proposed. Bamberger et al. proposed classifying FTMH as chronic if the patients reported a history of symptoms for over 2 years or a 1-year duration of FTMH after diagnosis [3].«

We understand your concern with the use of chronic and have therefore changed the concept from chronic to stage IV FTMH. The title and the manuscript have been changed accordingly.

Was this study approved by the ethics committee? If this is a prospective study, was the definition of chronic macular holes established at that time? If not, it would be better to restructure this as a retrospective study targeting chronic macular holes.

This was a retrospective study, approved by the ethics committee. We have corrected the manuscript accordingly.

“We conducted a retrospective study, enrolling patients with chronic stage IV FTMH, operated by a single senior consultant surgeon at the Department of Ophthalmology University Medical Centre Ljubljana, Ljubljana, Slovenia. The study was conducted according to the guidelines of the Declaration of Helsinki and approved by the National Medical Ethics Committee of the Republic of Slovenia (Approval No. 91/05/11).“

Please consider including the inclusion and exclusion criteria in the paper.

“Patients were enrolled from April 2021 to May 2023. Inclusion criteria were as follows: primary stage IV macular hole and pseudophakia. Exclusion criteria were preexisting ocular pathology (eg. Age related macular degeneration, glaucoma, high myopia (>-6), uveitis) and concomitant systemic comorbidities (eg. diabetes).”

Please provide more detailed information about the surgical procedure, such as: Was the ILM flap used to cover the macular hole or was it inserted into the hole? Were patients instructed to maintain prone positioning postoperatively, and if so, for how many days?

We have updated the manuscript as follows:

“At least two separate ILM flaps with their base connected to the macular hole margin were made during the ILM peel. Flaps were trimmed with a vitreous cutter and gently inverted over the hole and each other in a cabbage leaf-like manner (Figure 1, Video 1). As a rule, one flap was made temporally and the other nasally from the FTMH; occasionally we also made a third flap above or below the FTMH.”

The limitations of the study should be discussed, such as this being a case series and the lack of comparison with other surgical techniques.

Thank you for this suggestion; we have updated the manuscript accordingly.

»It is essential to recognize the limitations of this study. Firstly, the sample size is restricted due to the narrow inclusion criteria, focusing solely on patients with a stage IV macular hole who are also pseudophakic. Additionally, this study does not compare the cabbage leaf technique with alternative surgical approaches, such as the ILM peel or the inverted flap. Further research that includes comparisons among these techniques is warranted.”

Round 2

Reviewer 1 Report

Comments and Suggestions for Authors

Great work on reviewing your manuscript.

One last suggestion: there is excessive detail concerning macular hole classification. Maybe just present what stage IV is and how it is diagnosed.

Author Response

Thank you for your suggestion, we had to extend the word count of the manuscript per publisher's suggestion, so please understand we can not cut the words in the introduction section. 

Reviewer 2 Report

Comments and Suggestions for Authors

Thank you for the corrections. I believe the quality of the paper has improved.

Would it be better to include patients who underwent surgery with the cabbage leaf inverted ILM flap technique  in the inclusion criteria?

Author Response

We have updated the methods section as follows: 

Inclusion criteria were as follows: Patients with stage IV macular hole and pseudophakia, who underwent surgery with the cabbage leaf technique.